# Passive optical time-of-flight for non line-of-sight localization

Jeremy Boger-Lombard[1] & Ori Katz[1]

Optical imaging through diffusive, visually-opaque barriers and around corners is an important challenge in many fields, ranging from defense to medical applications. Recently, novel techniques that combine time-of-flight (TOF) measurements with computational reconstruction have allowed breakthrough imaging and tracking of objects hidden from view. These light detection and ranging (LiDAR)-based approaches require active short-pulsed illumination and ultrafast time-resolved detection. Here, bringing notions from passive radio detection and ranging (RADAR) and passive geophysical mapping approaches, we present an optical TOF technique that allows passive localization of light sources and reflective objects through diffusive barriers and around corners. Our approach retrieves TOF information from temporal cross-correlations of scattered light, via interferometry, providing temporal resolution that surpasses state-of-the-art ultrafast detectors by three orders of magnitude. While our passive approach is limited by signal-to-noise to relatively sparse scenes, we demonstrate passive localization of multiple white-light sources and reflective objects hidden from view using a simple setup.

[1] Department of Applied Physics, The Hebrew University of Jerusalem, Jerusalem 9190401, Israel. Correspondence and requests for materials should be addressed to O.K. (email: orik@mail.huji.ac.il)

In recent years there have been great advancements in techniques that enable non-line-of-sight (NLOS) optical imaging for a variety of applications, ranging from microscopic imaging through scattering tissue[1–8] to around-the-corner imaging[1,3,5,9–19]. The enabling principle behind these techniques is the use of scattered light for computational reconstruction of objects that are hidden from direct view. This has been achieved in a variety of approaches, such as wavefront shaping[3], inverse-problem solutions based on intensity-only imaging[18,19], speckle correlations[4,5,20], phase-space measurements[17], and time-resolved measurements[8–14]. While wavefront-shaping approaches allow diffraction-limited resolution, they require the presence of a guide-star at the target position[21] or long iterative optimization procedures. Inverse-problem solutions based on intensity-only imaging do not require prior mapping of the scattering barrier, but suffer from a drastically lower resolution, dictated by the diffusive halo blur dimensions. Phase-space based approaches[17], which localize hidden sources by measuring the positions and angles of the scattered light, suffer as well from a localization resolution that is dictated by the scattering angle of the barrier. Large scattering angles blur the k-space traces for objects that are not adjacent to the barrier.

A considerably higher resolution that is insensitive to the scattering angle of the barrier was recently obtained using speckle correlations[4–7] or by time-of-flight (TOF)[8–16] based approaches. The former rely on angular correlations of scattered light, known as the memory-effect[22], and allow diffraction-limited, single shot, passive imaging, using a simple setup. However, memory-effect based approaches suffer from an extremely limited field of view (FOV), and are limited to planar objects and to narrow spectral bandwidths. TOF-based approaches have recently allowed three-dimensional (3D) tracking and reconstruction of macroscopic scenes hidden from view[9,13,15,16]. These approaches utilize the principle of light detection and ranging (LiDAR) to obtain 3D spatial information from temporal measurements of reflected light that experienced several reflections (bounces). This is achieved by using ultrafast detectors to measure the time it takes a short light pulse to travel from an illumination point on the diffusive barrier, to the target object and back to the barrier. The scene is then computationally reconstructed from multiple TOF measurements at different spatial positions on the barrier.

Intuitively, one may assume that in order to measure time of flight, a pulsed source is mandatory, as is indeed used in most common LiDAR, radio detection and ranging (RADAR), and sound navigation and ranging (SONAR) systems. However, short-pulsed illumination is not a fundamental requirement: it is possible to obtain high-resolution temporal information from temporal cross-correlations of ambient broadband noise, without any active or controlled source. This principle is put to use in helioseismology[23], ultrasound[24], geophysics[25], passive RADAR[26], and recently in optical studies of complex media[27].

Here, we adapt these principles of passive temporal correlations-based imaging for 3D localization of hidden broadband light sources and reflective objects through diffusive barriers and around a corner. We retrieve high-resolution TOF information from scattered light using a simple, completely passive setup, based on a conventional camera. In our approach, temporal cross-correlations of scattered light are measured in a single shot, via low-coherence (white-light) interferometry, using controllable masks. Unlike conventional active TOF/LiDAR, where the temporal resolution is dictated by the pulse duration, or by the detectors' response time, in our approach the temporal resolution is dictated by the coherence time of the scene illumination. For natural white-light illumination, as used in our experiments, the TOF temporal resolution is approximately10 fs, three orders of magnitude better than the state of the art ultrafast

detectors[9,13,16]. However, while the localization resolution of our approach is superior to what is possible with ultrafast detectors, due to the passive nature of our approach and its reliance on low-coherence interferometry, it is fundamentally limited by signal-to-noise (SNR) considerations to the localization of a relatively small number of sources or reflectors, and it requires relatively long-exposure times. Unlike intensity-only[18,19] and phase-space measurements[17] approaches, the localization resolution in our approach is dictated by the TOF temporal resolution and not by the scattering angle of the barrier. We thus present localization results obtained through highly scattering diffusers, having a scattering angle of 80°, and using light scattered off a white-painted surface, using a simple, completely passive setup.

## Results

**Principle**. The principle of our approach is described in Fig. 1. Consider a small light source, or a reflective object hidden behind a diffusive barrier (Fig. 1a). For a source transmitting short pulses (or an object reflecting a short pulse), the source position can be determined by measuring the time of arrival of light from the source to different points on the barrier. Such TOF-based spatial localization is straightforward, as was recently demonstrated using ultrafast detectors[9,13,16]. However, when the illumination source is an uncontrolled continuous broadband noise source, such as natural white-light illumination, measuring the relative TOF is still surprisingly possible by temporally cross-correlating the random time-varying fields arriving at the barrier (Fig. 1b, c). Such passive correlation-imaging[28] is the underlying principle of our approach.

A numerical example for this principle is shown in Fig. 1a–c: the random time-varying fields from a broadband white-light source are measured at two positions on the barrier by two detectors (Fig. 1a). For a single hidden point source the measured fields arriving on the barrier, $E_1(t)$ and $E_2(t)$ are identical delayed versions of the source random field $E_s(t)$: $E_1(t) = E_s(t - \tau_1)$ and $E_2(t) = E_s(t - \tau_2)$ (Fig. 1b). Assuming free space propagation between the source and the barrier, $\tau_i = L_i/c$, where $L_i$ is the optical path length between the source and the $i$-th measurement point, and $c$ is the speed of light. The TOF difference $\Delta\tau = \Delta L/c = (L_1 - L_2)/c$, can be determined by temporal cross-correlating the two arriving random fields (Fig. 1c). For a sufficiently thin scattering barrier, the cross-correlation of the measured fields exiting the barrier is identical to the cross-correlation of the arriving fields, i.e., possessing a sharp peak with a temporal width that is equal to the coherence time of their illumination, centered at the relative time delay $\Delta\tau$ (see Fig. 1c and Supplementary Note 11). The source position can be determined in three-dimensions (3D) from three or more such TOF measurements taken at different points on the barrier, in a similar manner to the principle of global positioning system (GPS), and the recent NLOS imaging works in optics[9,13,16].

The spatial localization accuracy is dictated by the TOF temporal resolution, $\delta t$, i.e., by the temporal width of the cross-correlation peak. For a broadband source, this width is the source coherence time $\delta t \approx t_c \approx 1/\Delta f$, where $\Delta f$ is the source spectral bandwidth. This is easily shown by noting that the cross-correlation of the two fields, $E_1(t) \star E_2(t)$, is the autocorrelation of the source field, shifted by $\Delta\tau$:

$$E_1(t) \star E_2(t) = (E_s \star E_s)(t - \Delta\tau) \qquad (1)$$

For a broadband source, the autocorrelation $(E_s \star E_s)(t)$, is a narrow, sharply peaked function around $t = 0$, with a temporal width that is given by $\delta t \approx 1/\Delta f$ according to the Wiener-Khinchin theorem. Thus, the fields cross-correlation would display a sharp peak at the time delay $t = \Delta\tau = (L_1 - L_2)/c$, providing the TOF

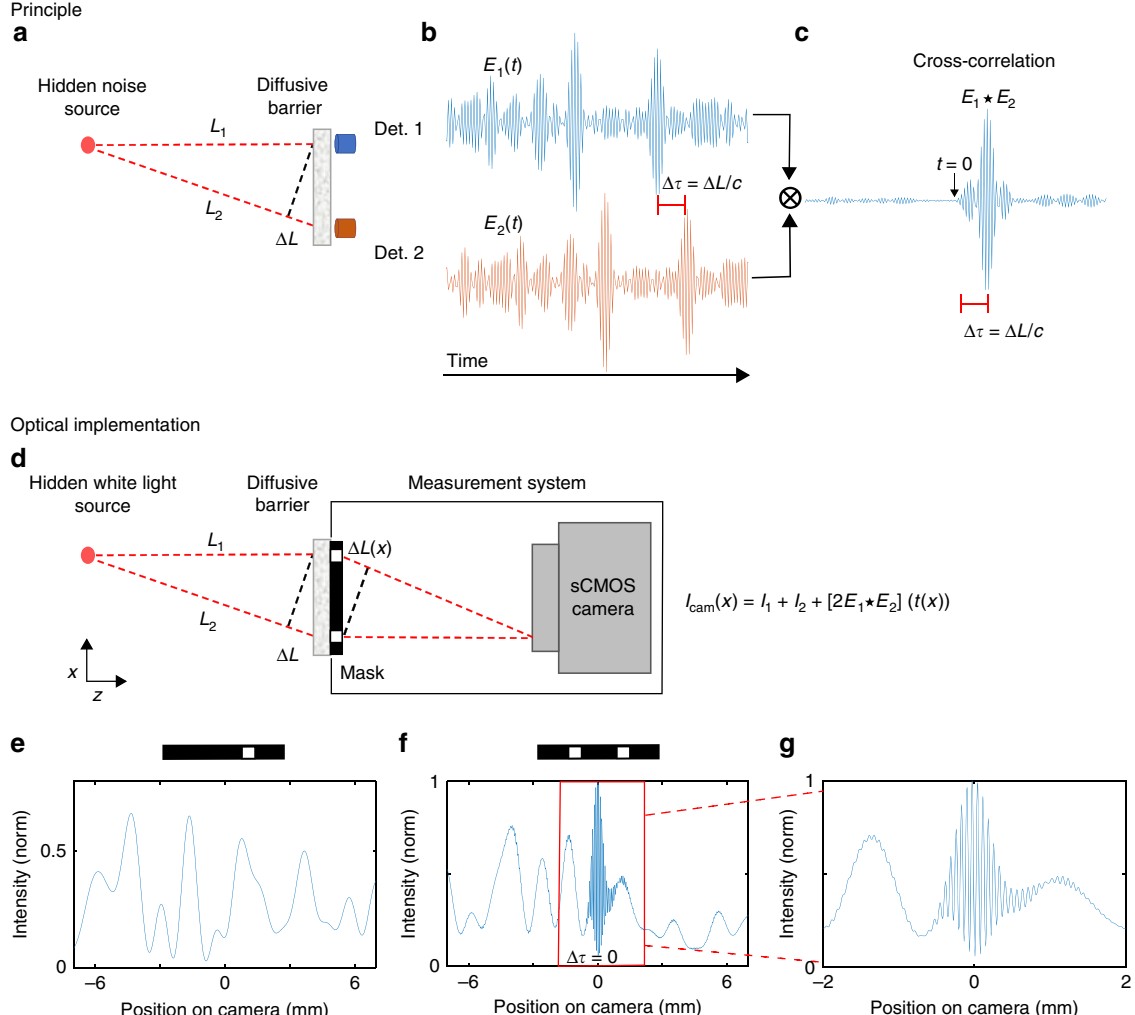

**Fig. 1** Passive TOF by temporal cross-correlations. **a**–**c** Principle: a white-light source or a reflective object is hidden behind a diffusive barrier. **a** The random noise field from the source is measured at two points on the barrier by detectors 1 (blue) and 2 (orange). **b** For a thin barrier, the measured fields are replicas of the same broadband noise, shifted by the time-of-flight (TOF) difference $\Delta\tau = \Delta L/c$. **c** Cross-correlating the measured fields reveals the TOF difference as a sharp cross-correlation peak. **d** Optical implementation: light from two points on the barrier is selected by a controllable mask and interferes on a high-resolution camera. A single long-exposure camera shot $I_{cam}(x)$ provides the fields temporal cross-correlation at different time delays, $t(x) = \Delta L(x)/c$. **e**–**g** Numerical example of the measured intensity patterns on a single camera row. **e** For a mask with a single aperture, a speckle intensity pattern is measured. **f** For a double-aperture mask, a cross-correlation peak appears as white-light interference fringes on top of the speckle pattern, providing the TOF difference $\Delta\tau$. **g** Zoom-in on (**f**), the TOF resolution is the source coherence time

difference from the hidden source to the two points on the barrier (Fig. 1c). Such a single TOF difference measurement localizes the source to be on a hyperboloid surface, with its foci at the two detectors. Repeating this passive TOF measurement at two additional positions on the barrier would localize the source in 3D.

Direct implementation of such a field correlation approach using two optical detectors is not straightforward since measuring the temporal variations in optical fields requires phase-sensitive ultrafast detection with sub-femtosecond temporal response. Conventional optical detectors measure only the optical intensity averaged over response times that are several orders of magnitude longer. However, access to the optical fields' temporal cross-correlation is directly possible via interference, even with very slow detectors. Our approach utilizes this principle to implement passive optical TOF measurements, as presented in Fig. 1d–g and as explained below.

Figure 1d depicts a simplified optical setup that enables spatially resolved temporal cross-correlation measurements via

interferometry, using a conventional high-resolution camera. First, the light from two chosen positions on the barrier is selected by a mask having two apertures. The mask is placed adjacent to the barrier or imaged on its surface. The light fields passing through the two apertures, $E_1(t)$ and $E_2(t)$, interfere on a camera detector placed at a sufficiently large distance from the mask. In a single long-exposure image, each camera pixel intensity value, $I_{cam}(x)$, at a transverse position $x$, is the time-integrated optical intensity resulting from the interference of the two fields:

$$
\begin{aligned}
I_{cam}(x) &= \int_{-\infty}^{\infty} [E_1(t) + E_2(t + t(x))]^2 \, dt \\
&= I_1 + I_2 + 2[E_1 \star E_2](t(x))
\end{aligned}
\tag{2}
$$

Where $I_j$ is the time-averaged intensity that passes through the $j$th aperture. Thus, a single camera row provides the fields temporal cross-correlation, $E_1 \star E_2$, sampled at thousands of different time delays, given by the positions of the camera pixels, $x$, at that row: $t(x) = \Delta L(x)/c$. The sampled time delays are dictated by the system's geometry (see Supplementary Fig. 6). Using a single

aperture pair, the number of time delays that can be sampled in a single exposure is limited by the number of pixels in a single camera row, which was 2160 in our experiments.

The source TOF difference can be easily determined from the position of the cross-correlation peak, manifested as low-coherence interference fringes (Fig. 1f, g). This bears similarity to white light interferometry[29] and optical coherence tomography (OCT)[30]. However, in contrast to these techniques, in our approach no reference arm or external source are used, and the measurements are self-referenced, i.e., cross-correlated.

In practice, for a strongly scattering barrier, the light intensity from each of the apertures on the mask produces a spatially varying random speckle intensity pattern on the camera pixels (Fig. 1e). However, the speckle intensity patterns do not mask the low-coherence interference fringes, as we design the fringes period to be considerably smaller than the speckle grain size (Figs. 1f, g and 2b, c).

**3D passive TOF measurement through a diffusive barrier.** Figure 2 presents experimental results of passive TOF measurement using a setup based on the design of Fig. 1d. The full experimental setup is given in Supplementary Fig. 1. As a first demonstration, a single small white-light light-emitting diode (LED) source was hidden behind a highly scattering diffusive barrier, having an 80° scattering angle and no measurable ballistic component (80° light shaping diffuser, Newport). In this experiment, a movable mask, comprised of four small apertures (Fig. 2d), is used to interfere light from two pairs of points on the barrier. The points are vertically and horizontally spaced to obtain TOF information on both elevation and transverse position of the hidden source, respectively, in a single camera shot.

When a mask with only a single aperture is used, the camera image is a random speckle pattern (Fig. 2a). This random speckle pattern provides no useful spatial information on the source position. However, when a mask with two horizontally spaced apertures is used, low-coherence interference fringes appear on top of the random speckle pattern at a specific horizontal position on the camera image (Fig. 2b, c). This white-light fringe pattern is the result of the interference between the speckle patterns that are transmitted through each of the two apertures. Since the hidden source is a broadband white-light source, the interference fringes are located only around the zero path delay difference, i.e., when the path difference accumulated after the diffuser $\Delta L(x)$ is equal to the path difference accumulated before the diffuser: $\Delta L(x) = \Delta L$, where $x$ is the camera pixel transverse position (Fig. 1d), similar to Young double-slit experiment with low-coherence light. Thus, the optical path length (or TOF) difference of the light from the source to each of the two apertures can be extracted from the fringes position present on a single camera image (Fig. 2c), with a resolution given by the coherence-length (or coherence time) of the source. The practical TOF resolution is also limited by any additional path delay spread inside the diffusive barrier (see Discussion section, and Supplementary Note 11).

A single TOF difference measured from a single camera image, localizes the source position to be on a hyperboloid surface. To localize the source to a single point, additional measurements are required. Two additional measurements with the mask shifted to two different positions will provide two additional hyperboloids. The intersection of the three hyperboloids can localize a single source to a single point in 3D.

Using more complex masks can reduce the required number of camera shots. Inspired by aperture masking interferometry[31,32], we designed a mask that multiplex two TOF measurements in a single camera shot. This mask, presented in Fig. 2d, comprises two pairs of apertures, simultaneously providing two TOF measurements by generating both vertical and horizontal fringes. Figure 2g presents an example for a raw camera image acquired using this mask.

The position of the horizontal and vertical high spatial-frequency fringes can be easily extracted using spatial bandpass filtering and a Hilbert transform of the camera image (or alternatively by a spectrogram analysis of the camera image rows and columns, as detailed in Supplementary Fig. 3). The results of such filtering on the camera image of Fig. 2g are shown in Fig. 2j, m, revealing the vertical and horizontal fringes amplitude respectively. The additional required TOF measurements are obtained by shifting the mask position, horizontally (Fig. 2e, h), and/or vertically (Fig. 2f, i). When the mask is horizontally shifted only the vertical fringes position changes (Fig. 2k), while the horizontal fringes position remains fixed, as expected (Fig. 2n). When the mask is vertically shifted, the horizontal fringes shift upward (Fig. 2o), and the vertical ones remain fixed (Fig. 2l).

**Localization of multiple light sources.** In the case of multiple hidden sources, each single camera image contains several interference fringe patterns at different positions. Each fringe pattern position provides the TOF for a single hidden source. For independent spatially incoherent sources, such as natural light sources, the number of interference fringe patterns is equal to the number of hidden sources, as the light from different sources does not interfere.

As in active TOF-based NLOS imaging approaches, localizing multiple hidden sources without ambiguity requires a larger number of TOF measurements. This can be achieved by shifting a single mask to multiple positions and measuring the TOF difference for each mask position. An experimental example for such localization of two and three hidden sources in two dimensions is shown in Fig. 3. Figure 3b displays the measured fringes' amplitude envelope at 40 different mask positions: each row in Fig. 3b displays the fringe amplitude extracted from a single camera image (see Supplementary Fig. 3), where the horizontal axis is the TOF delay (camera pixel), and the vertical axis is the mask position. Inspection of Fig. 3b clearly reveals that two hidden sources are present at the hidden scene.

Each intensity-peak in each row in Fig. 3b, localizes the sources to a hyperboloid. The sources are unambiguously localized from the intersection of multiple back-projected hyperboloids (see Supplementary Note 4), as is demonstrated in Fig. 3c, d.

In the experimental results of Fig. 3, the mask position was mechanically scanned. A similar acquisition can be performed without any moving parts, by replacing the mask with a computer-controlled spatial light modulator (SLM). An experimental demonstration using such a programmable mask for passive TOF measurements is presented in Supplementary Fig. 2. The advantages of an SLM-based mask are its versatility and speed, in particular for advanced multiplexing using complex multi-apertures masks (see Supplementary Note 2).

**Localization around a corner.** Our passive TOF approach can be used to localize sources around a corner, using light that was scattered off diffusive white-painted walls[9,13,16]. Figure 4 presents a proof of principle experiment for around the corner passive localization. Figure 4a shows the experimental arrangement, which is conceptually identical to the design of Fig. 2a, with the only difference that the diffusive barrier is replaced by a white-painted wall, and that the wall's surface is imaged on the movable mask.

In order to localize multiple light sources that are placed around the corner, the same measurement protocol described above (Fig. 3) was performed. The measured spatio-temporal ($x$–$t$) trace of fringes' position on camera (i.e., TOF, $t$) vs. mask position ($x$) is shown in Fig. 4d. The scene reconstructed via back-projection is shown in Fig. 4e.

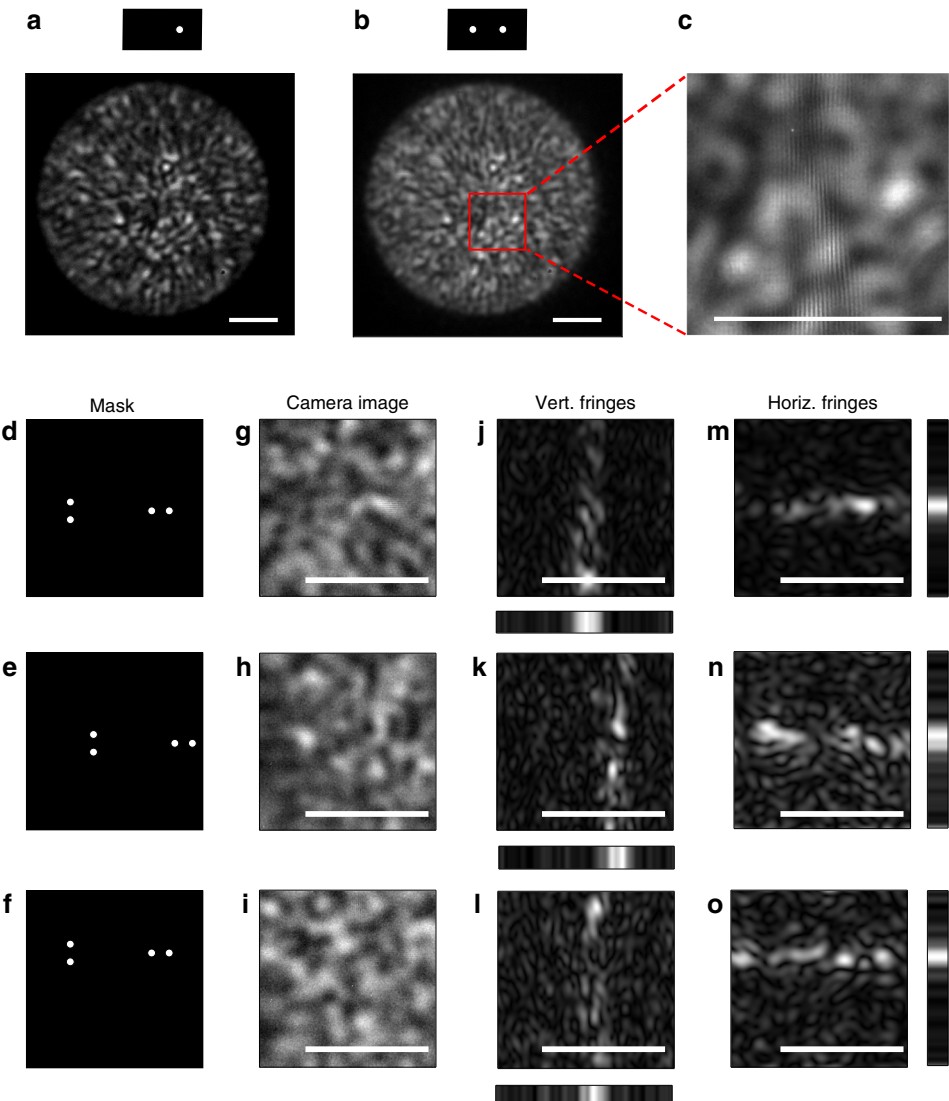

**Fig. 2** Hidden source experimental TOF measurement through a diffusive barrier. **a–c** Raw camera images showing: **a** The speckle pattern measured for a mask with a single aperture (top). **b** For a double-aperture mask (top), localized interference fringes (highlighted by a red square) provide the TOF difference from the hidden source to the two apertures. **c** Zoom-in on (**b**). **d–o** Multiplexing elevation and horizontal information is accomplished using a mask with two pairs of apertures. 3D localization is obtained by shifting the mask for multiple TOF measurements: **d–f** Shifting the mask horizontally (**e**) and vertically (**f**) as imaged on the barrier. **g–i** Raw camera images for each mask position. **j–l** Vertical fringes envelope amplitude as detected by filtered Hilbert transform, providing the TOF difference between the two horizontally separated apertures (bottom: sum over rows). **m–o** Same as (**j–l**) for the horizontal fringes envelope amplitude (right: sum over columns). Scale bars: 100 fs

As can be observed in Fig. 4d, the TOF traces from the reflective wall display TOF fluctuations that are larger than those observed through the optical diffuser (Fig. 3b). This effect is a result of the rough nature of the white-painted wall. It can be observed and quantified thanks to the unique femtosecond-scale temporal resolution of our approach, whereas picosecond pulses used in previous works would mask these effects. To show that the barrier roughness indeed induces measurable variations in the TOF with our approach, we calculated the standard-deviations of the measured TOF for both the diffusive barrier in transmission and the white-painted wall in reflection to be ~4.6 fs, and ~20 fs, respectively (see Supplementary Fig. 12).

The surface roughness and the multiple-scattering nature of the reflections from the white-painted barrier, induce not only fluctuations in the TOF differences, but also a temporally broader impulse response. This temporal broadening is also measurable by our system, as is shown in Fig. 4f. This figure presents a

comparison of the fringes envelope as a function of the time delay for the case of a thin diffuser vs. the white-painted barrier. Compared with the diffuser case, for the white-painted wall we observe a correlation peak that is broader by a factor of ~2, accompanied by a broader pedestal. This broadening, in addition to the TOF fluctuations due to the surface roughness, effectively lower the TOF resolution and reduces the localization accuracy. Nonetheless, the TOF resolution is still generally better than using picosecond pulses, and our approach is still able to successfully localize the source positions in reflection in our proof-of-principle experiments.

Interestingly, the changes in the TOF curves that are induced due the nature of the barrier, and are resolvable with our system, carry information on the barrier. For sufficiently small point-like hidden sources, the surface and scattering properties of the barrier, such as its transport mean free path and/or roughness, can in principle be estimated from the temporal

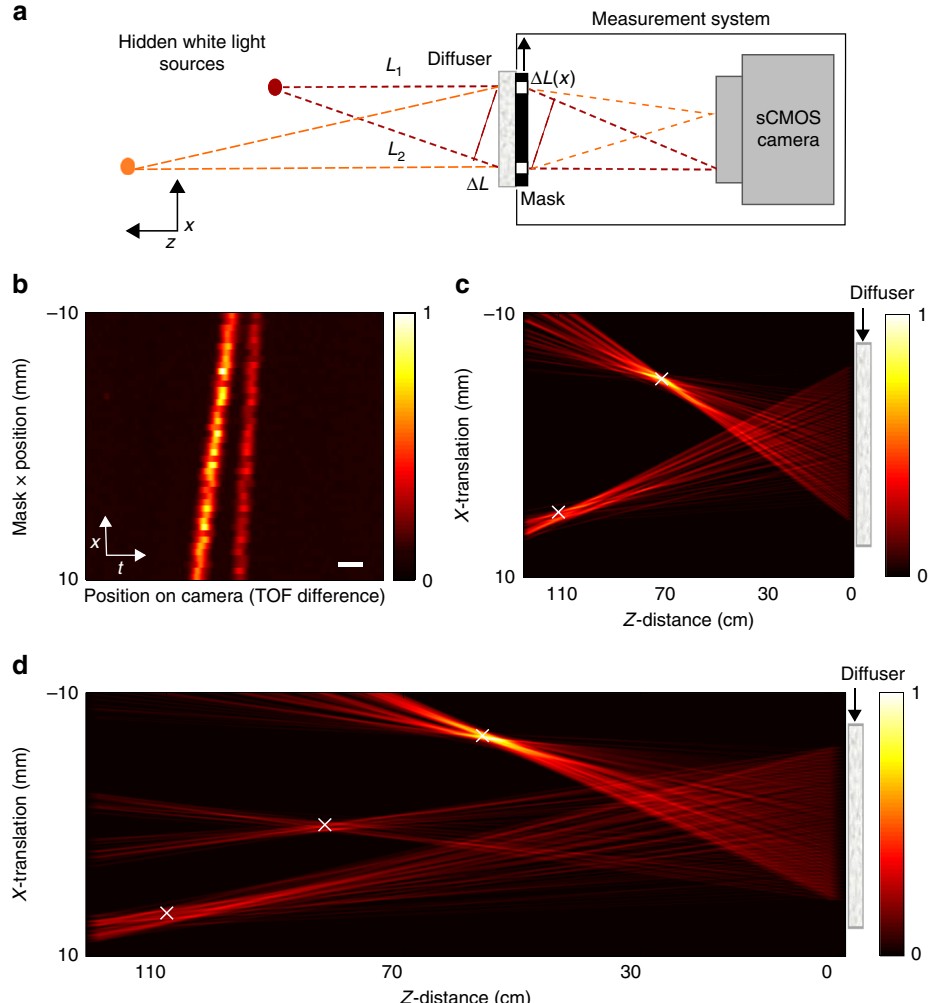

**Fig. 3** Experimental localization of multiple hidden sources. **a** Simplified sketch of the setup with two hidden sources (white-light LEDs). **b** Detected envelope of the interference fringes as a function of the mask position (vertical axis). The fringes positions (horizontal axis) mark the TOF differences. **c** Hidden scene reconstructed from (**b**) by back-projection. Each fringe position in a row in (**b**) localizes the sources to a hyperboloid. The sources real positions are marked by 'x'. **d** Back-projection reconstruction of a scene containing three hidden sources. Scale bar: 100 fs

broadening and fluctuations of the low-coherence fringes envelope[27].

**Localization of reflective objects**. The approach can also be used to localize small reflective objects. Figure 5 presents such a demonstration using the same experimental system, with the only difference that two small metallic objects are placed in the scene, and the dark-background scene is illuminated by a halogen lamp (Thorlabs OSL2). The measured spatio-temporal ($x$–$t$) TOF trace and scene reconstruction are presented in Fig. 5b, c.

**Discussion**
We have demonstrated an approach that allows to passively localize incoherent light sources and reflective objects through diffusive barriers and around corners using standard cameras. As in conventional (active) TOF approaches[9,13,16], the spatial localization resolution is determined by the TOF temporal resolution and the setup's geometry. However, unlike conventional TOF approaches, in our approach the TOF resolution is given by the light source coherence time, rather than the detectors response time. For the white-light sources used in our experiments, the coherence time is of the order of 10 fs (see Supplementary Note 5), more than three orders of magnitude better than the

response time of state-of-the-art streak-cameras or single-photon avalanche diode (SPAD) detectors[9,13,16]. This temporal resolution may be an advantage in, e.g., microscopic imaging applications. However, as we discuss in detail below, while the localization resolution is better than active techniques, the passive nature of our approach and its low-coherence interferometry based measurements fundamentally limit its use to localizing a relatively small number of small sources or reflectors, a limitation that does not occur in active TOF-based approaches, where the measurement background is low compared with the active pulsed illumination intensity.

Beyond the TOF resolution, the main advantage of our approach is in it being passive and having a simple implementation, not requiring fast detectors or streak cameras. Our technique makes use of natural incoherent light, present in many natural scenes, in a fundamentally new fashion.

The passive nature of our TOF approach is also its main disadvantage: since light from natural light sources is considered, the detected intensity levels are inherently low. In our implementation, integration times of several seconds for a single camera shot were required to provide sufficiently high contrast fringes from the white-light LED sources in the diffusive barrier case (Fig. 3), under dark conditions. In the experiments with reflective objects (Fig. 5),

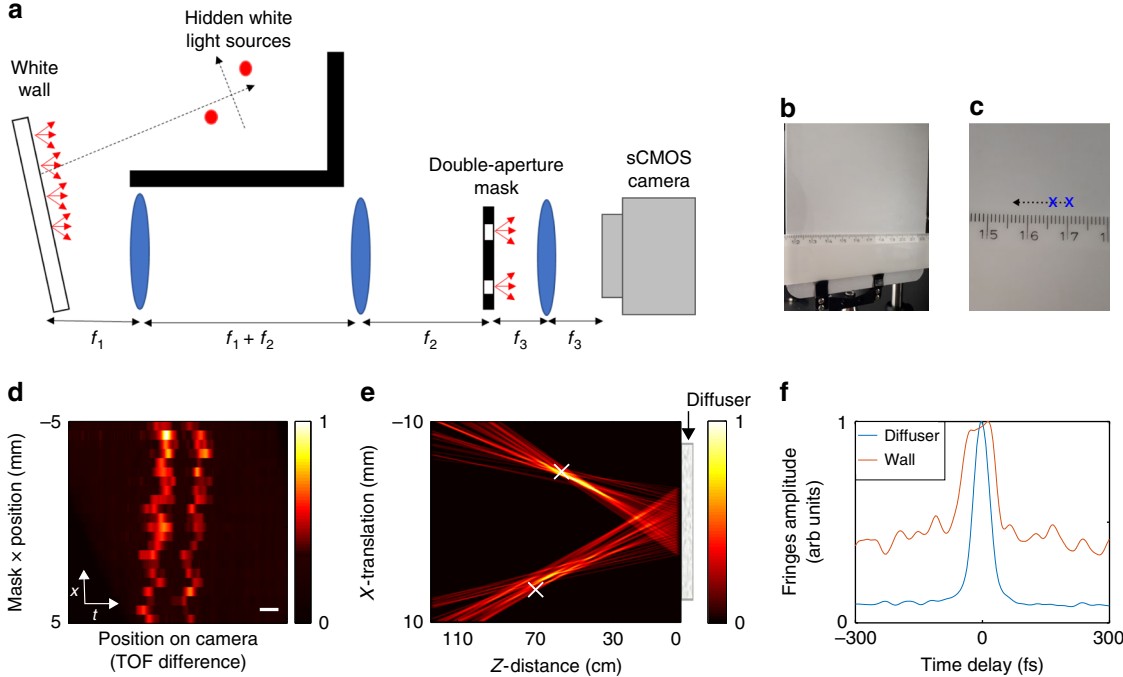

**Fig. 4** Localizing multiple light sources around a corner. **a** Setup (top view): hidden white light sources illuminate a white-painted surface. The wall surface is imaged by a 4-f telescope on a movable double-aperture mask. The light from the two apertures is interfered on the camera by a lens. **b**, **c** Photographs of the wall surface. The imaged apertures positions are marked by 'x'. The mask scan trajectory is depicted by a dashed line. **d** Interference fringes envelope as a function of the double-aperture mask position, revealing the two hidden sources. **e** Scene reconstructed from (**d**) by back-projection. The sources real positions are marked by 'x'. **f** fringes envelope as a function of the time delay, for the case of a thin diffuser (blue), and the white-painted barrier (orange). Scale bar: 100 fs

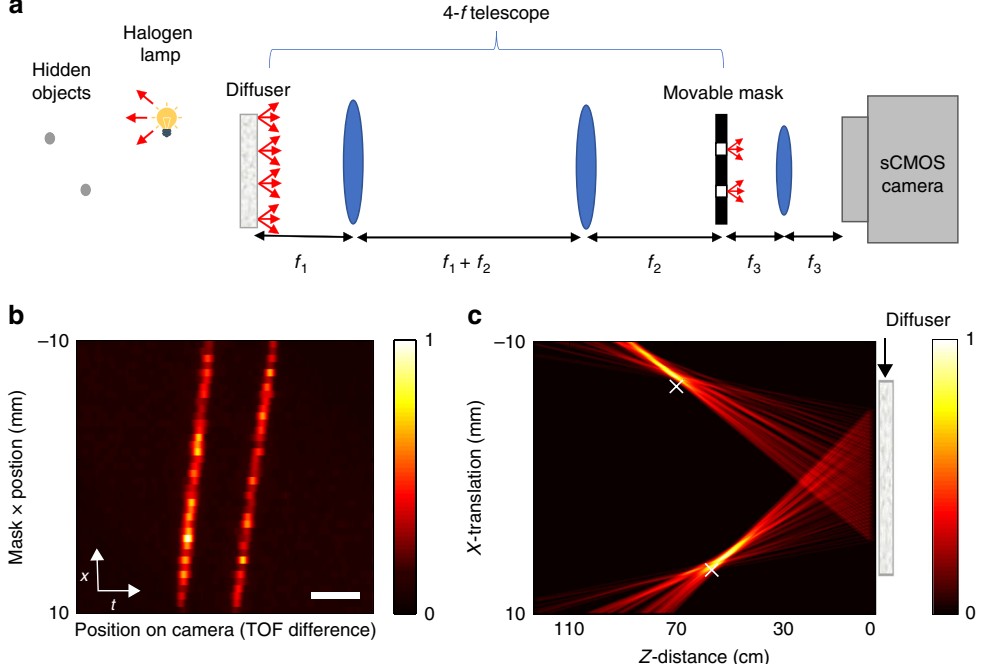

**Fig. 5** Localizing hidden reflective objects. **a** Setup (top view): two hidden metallic objects are illuminated by a halogen lamp (Thorlabs OSL2). The reflected light passing through a highly scattering diffuser is interfered on a camera by a movable double-aperture mask. **b** Interference fringes envelope as a function of the mask position, providing the TOF information. **c** Scene reconstructed from (**b**) revealing the sources positions. The objects' real positions are marked by 'x'. Scale bar: 100 fs

exposure times of 30 s were used for a single exposure. In the around-the-corner experiments, exposure times of 15 min per camera shot were required (Fig. 4). These single-shot exposure times yielded total acquisition times of minutes to hours for the total number of measurements presented. Lower exposure times lead to lower fringe visibility and signal-to-noise ratio (SNR) but can still be useful for TOF measurements, as we study in Supplementary Note 8. A lower number of measurements can still be used for localization at a lower accuracy (see Supplementary Fig. 9). Reduced acquisition times and number of acquisitions may be possible by using multiplexed detection with masks having a large number of apertures, and by using more advanced reconstruction algorithms. In our proof-of-principle experiments we have chosen to use simple masks comprised of two groups of circular-apertures pairs to multiplex two TOF measurements in a single camera exposure. More complex masks can be used to allow scene reconstruction from a single camera exposure, in a similar manner as is done in aperture masking interferometry in astronomy[32].

To achieve a sufficiently high fringe contrast, the experiments presented in Figs. 2–5 were performed under dark conditions. The effects of ambient light and bright reflecting background on the fringe contrast are studied in Supplementary Note 8. For the hidden white-light LED sources similar localization results can be obtained under both dark and ambient-light conditions. For the reflective objects case, the quality of the results depend on the relative illumination intensity of the objects compared with the background. The requirement for bright sources or reflectors over a relatively dark background is a fundamental requirement for obtaining high contrast fringes. However, as in OCT, low contrast fringes may be resolved by longer exposure times, achieved by averaging multiple exposures or by large well-depth cameras.

Similar to other TOF-based NLOS imaging approaches, the localization resolution is determined by the geometry of the system and the TOF temporal measurement resolution. The transverse ($dx$) and axial (depth, $dz$) localization resolution from a single camera shot in our approach are analyzed theoretically in Supplementary Note 6. They are given by $dx \approx l_c \cdot z/D$ and $dz \approx \frac{l_c \cdot z^2}{D(x + x_{slits})}$ (Supplementary Eqs. (9) and (8) respectively), where $x$ is the source\object transverse position in respect to the center of the measurement, $x_{slits}$ is the center position of the moving slits, $l_c$ is the coherence length of the light source and $D$ is the slits separation . Substituting our experimental parameters of $l_c = 6.3$ μm, $x = 0$, $x_{slits} = 10$ mm, and $z = 80$ cm, yields: $dz \approx 32$ cm and $dx \approx 4$ mm, for the single-shot localization resolution. In order to verify our theoretical predictions for the localization resolution, we have performed a set of experiments with two sources separated by various transverse and axial distances. The results of these experiments, which validate our theoretical estimates, are presented in Supplementary Note 6.

The final localization accuracy is considerably better than the single-shot localization resolution, since it is obtained from multiple TOF measurements. An experimental study of the localization accuracy as a function of the number of TOF measurements is presented in Supplementary Note 7.

The theoretical localization resolution improves for larger apertures separation, $D$. However, when extended objects are concerned, the aperture separation must be smaller than the source coherence size on the barrier, $r_{coh}$, to obtain high contrast fringes. This limits the apertures separation when large extended sources or objects are considered, and limits the single-shot localization resolution to the source dimensions (see Supplementary Note 6). Improved localization accuracy can be obtained by scanning the double-aperture mask over larger distances ($x_{slits}$).

In our experiments with thin diffusive barriers and highly scattering walls, the fringe visibility was high. However, when transmission through thick diffusive barriers is considered, the fringe visibility, as well as speckle contrast, will be reduced due to the narrower speckle spectral correlation width, resulting from the larger spread in optical paths[33,34] (see Supplementary Notes 9–11).

The presented technique is conceptually similar to Michelson stellar interferometry, where the fringe contrast (visibility) as a function of the apertures separation is used to reconstruct the shape of a distant star, as was also recently demonstrated for NLOS object reconstruction[35]. However, in our approach the apertures separation is fixed, and only the positions of the low-coherence fringes positions are used to extract the TOF information. Our approach can be combined with such stellar interferometry inspired approaches to reconstruct also the hidden source's shape, and not just its position. Interestingly, the same information that was obtained in our approach via. interferometry can in principle be obtained from intensity-only correlations using the Hanbury Brown and Twiss (HBT) approach[36].

In our approach, differences in TOF are measured and not the total round-trip TOF, which is measured in active TOF approaches[9,13,16]. This leads to localization of the hidden sources on spatial hyperboloids in 3D, rather than ellipsoids[9,13] or spheres[16] of previous works.

As in recent NLOS works, we have chosen to reconstruct the complex scenes via back-projection. Alternatively, when single-localized sources are present, a single source position can also be determined from the slope and position of its high brightness spatio-temporal measured curve (see, e.g., Fig. 3b), in a similar manner to the principle of parallax-based localization, which was also recently used for localization based on phase-space measurements[17].

Unlike memory-effect based works[5], the FOV of our technique is not limited by the memory-effect, since each source position is measured independently. However, for our technique to work well through thick diffusive barriers, the apertures separation, $D_{slits}$, must be larger than the thickness of the barrier (in transmission) or its transport mean free path (in reflection), to minimize mixing between the signals in two apertures, induced by the light spreading inside the barrier.

## Methods

**Experiments**. The full experimental setups are presented in detail in Supplementary Fig. 1. The hidden light sources were generated by splitting the light from a white-light LED (Thorlabs MWWHF1) to four, using a fiber bundle (Thorlabs BF42HS01), effectively producing white light sources of 200-μm diameter and numerical aperture of $NA = 0.39$, having an average power of ~1.5 mW. The diffusive barrier was a Newport light shaping diffuser with a scattering angle of 80° (40° for the experiments with reflective objects (Fig. 5)) and a transmission of >90% in the visible range.

The scattering wall was a metal plate painted with white matte spray (Tambour 465-024). The light sources were placed at various positions with distances of 30–110 cm from the diffuser and 56–70 cm from the wall. For the object localization experiment the light source used was a white light halogen lamp (Thorlabs OSL2) culminated with its culmination package (Thorlabs OSL2COL). The object was a metallic nut covered with black tape leaving a reflective area 3 mm high and 0.7-mm wide. The diffusive barrier/wall was imaged on the aperture masks using a 4-f telescope. The light collection aperture diameter on the first lens was 5 cm. For the measurements shown in Figs. 2 and 5 and for around the corner measurements, the aperture was limited to 2.5 cm. The apertures masks were fabricated by drilling 0.25-mm-diameter circular holes in ~2 mm-thick black Delrin plates. The separation between the apertures was 3.2 mm. For the experiments with reflective objects (Fig. 5) and the system's resolution and accuracy analysis measurements (Supplementary Figs. 5, 7, 8 and 10) a mask with 0.6 mm diameter holes and separation of 1.5 mm was used. The light passing through the mask was focused on an sCMOS camera (Andor Neo 5.5) with an $f = 10$ cm lens in an f–f configuration (see Supplementary Fig. 1). Each TOF measurement was acquired by a single long-exposure (15 s to 15 min) image.

## Data availability
All relevant data are available from the authors upon request.

## Code availability
All relevant codes are available from the authors upon request.

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

## Acknowledgements
This material is based upon work supported by the Defense Advanced Research Projects Agency (DARPA) under Contract No. HR0011-16-C-0027, the ISRAEL SCIENCE FOUNDATION (Grant no. 1361/18), and the European Research Council (ERC) under the European Union Horizon 2020 research and innovation program (Grants no. 677909). O.K. acknowledges support from the Azrieli Faculty Fellowship, Azrieli Foundation. The information presented in this work does not necessarily reflect the position or policy of either DARPA or the U.S. Government, and no such official endorsement should be inferred.

## Author contributions
O.K. conceived the idea, J.B.L. and O.K. designed the experiments and performed numerical simulations. J.B.L. conducted the experiments and analyzed the data. J.B.L. and O.K. wrote the paper.

## Additional information

**Competing interests:** The authors declare no competing interests.

