## [Peer Review File · Nature Communications]

Recently, numerous studies pushed back the limits of non-line of sight (NLOS) imaging techniques. Usually, these techniques require pulsed sources and (very) high-speed photo-detectors in order to measure time of flight (ToF) of the order of picoseconds. This results in complex and expensive set up. In this article, the authors propose to adopt an approach already established in seismology, acoustics and more recently in optics to perform similar task, here source localization, but in a totally passive manner. Passive means no pulsed source or high speed-detectors are needed. This approach relies on the interferences of low coherence light emitted from the sources and measured with a conventional camera. Furthermore, the resolution is no longer dictated by the detectors but by the coherence time of the sources which can be in the femtosecond range for white light lamps.

I found this paper very interesting. It is well written and presents an elegant approach to simplify current techniques. The experimental setup is rather simple, cost-effective and the experimental demonstrations are convincing, especially as different configurations are explored. The supplementary information is also valuable. Maybe too much, as I found it really necessary to understand the approach (which is not that simple with the article only). In my opinion, this work is of interest for the community but the article in his current version lacks some clarifications on possible applications, more information on the experimental conditions and details on the limitations of this technique. Here is the list of my comments and concerns:

- (1) First, the two advantages of the proposed technique are the passive nature of the measurement and its temporal resolution. In NLOS imaging techniques, this temporal resolution is directly related to the spatial resolution by the celerity of light. A better temporal resolution, here the authors claim temporal resolution 3 order of magnitude better than conventional techniques using ultrafast detectors, should then provide higher spatial resolution. This is not the case in this article where the set up's geometry limits the spatial resolution as detailed in the supplementary information only.

It seems necessary to me to clearly describe what is the ideal spatial resolution (from the temporal resolution) and how the setup reduces it. Currently, the authors provide no number to characterize the spatial resolution of the system, they just mention "the object angular size", which is not practical at all.

For instance, I would like to know, how close can 2 sources be localized in the current configuration? And more important, how can it been improved?

Right now, the results show sources separated by tens of centimeters, so way larger compared to conventional NLOS techniques where the spatial resolution seems to be in the millimeter-centimeter range. Can this method achieves similar performances? Or is it the best performances achievable with this technique?

- (2) My second question is also about the localization accuracy. Theoretically, 3 ToF measurements only are needed to estimate the position of a source. Here, more ToF are used which of course improve the accuracy (and improves the appearance

of the experimental data). I think it could be interesting to display a plot of the accuracy as a function of the number of ToF.

- (3) The authors mentioned only once the exposure time. Are the values (few seconds for the diffusive barrier and 15 minutes for the around the corner experiment) for one ToF measurement only or for the total acquisition? I think it should be mentioned for each experiment.
- (4) Also, what is the acquisition scheme? Is it a single frame with long exposure or multiple images with low exposure averaged in post processing? I think adding a convergence curve (visibility of fringes as a function of exposure time) and examples of images with different exposure time would be really useful.
- (5) The authors state the cross correlation is sampled at thousands of different time delays (line 111). Can they be more specific? The camera has 5.5 Mpixels, so do they use the full resolution? Maybe the authors can state the number of sampling points is given by the number of pixels (if I understood well), and then give the number of pixels used in their experimental conditions.
- (6) Are the experiments performed with a background light? Is this approach doable outside for instance?
- (7) What is the transmission efficiency of the diffuser and of the mask?
- (8) Why the authors choose this particular mask? With only 2 groups of 2 square elements? Why the authors did not choose rounded apertures for instance? And why not groups of 3 apertures (in a triangle form) to provide both x and z as the same time?
- (9) In the experiment around the corner, the ToF results display both a broader and a rougher response. From the broadening, is it possible to estimate the scattering properties of the wall? Does the rough aspect comes from the rough surface of the wall or just from lower signal to noise ratio? Would it be possible to estimate other parameters on the wall from the roughness of the ToF?
- (10) The two lamps used in the experiments have different spectral bandwidths. Do the authors noticed different temporal resolutions in the experiments conducted with these two light sources?

Minor comments

- a) The references line 22 should be splitted in 2 and ordered by date.

b) Line 235 : there is a comma after star

Reviewer #2 (Remarks to the Author):

This paper describes a technique for localizing a single or a few points through a thin diffuser or "around a corner". What's unique about this approach is that, unlike recent work in this important and timely research area, it does not rely on pulsed illumination and ultra-fast detectors nor does it require coherent illumination; it is therefore a passive imaging approach to non-line-of-sight imaging. Several other passive techniques to this problem have been proposed over the last few years, as adequately discussed in the manuscript, but the proposed method uses interference created by a "coded aperture" (pairs of horizontal or vertical points that are mechanically scanned on the diffuser plane) to achieve substantially higher resolution for the localization problem.

The idea behind the proposed technique is intuitive and closely related to Young's double slit experiment: the light emitted or reflected by a point source is constrained by only allowing it to pass through two small points or slits. After propagating by some distance, an interference pattern is created on the sensor. Point sources emitting light from different directions on the points/slits create a phase shift on the slit plane that is proportional to the incident direction, which in turn shifts the interference pattern. While a single image of the interference pattern is insufficient to localize the direction of the source, two approaches to using this effect for non-line-of-sight imaging seem possible. First, one could track a moving point by observing the shift of the interference pattern (not discussed in this paper) or (2) one could vary the distance between the slits for a stationary point source (proposed in this manuscript). This authors introduce the latter concept for the application of non-line-of-sight imaging and experimentally validate it for several conditions, including a single and multiple point sources, points behind a diffuser, and points scattered off of a reflective wall.

Overall, this is a very clever and intuitive technique, the manuscript is well written, and the experimental results are convincing in that they demonstrate that this intuitive concept is indeed applicable to source localization in non-line-of-sight imaging scenarios.

Reviewer #3 (Remarks to the Author):

The paper describes a method for localizing a light sources obscured by diffusers using an interferometric technique. The paper is well written and does a good job exploring different applications of this method. I have only a few comments:

1. Reference [30] seems to describe a related method, but is only ever cited in passing. A more detailed comparison to this work in the introduction is warranted here as it seems on first sight to be the work that is closest related to this manuscript.

2. In computational imaging "passive" techniques are usually those that work with ambient light, while "active" techniques require control of the illumination, for example from a projector. Using this definition, the presented method is active since it requires the targets to light up and presumably does not tolerate ambient light. Even the setup described in Figure (a) apparently uses illumination that is aimed only at the targets. I would recommend replacing the term Passive by something less confusing. Maybe 'coherence gated', 'interferometric', or 'time-of-flight with conventional cameras'

Overall I recommend publication of the manuscript.

Please find attached below our point-by-point to each of the referees comments.

We sincerely thank the referees for their insightful comments and suggestions that allowed us to significantly improve our manuscript. In order to properly address all of the referees' comments and suggestions, we have performed additional experiments and analyses, which include the theoretical and experimental quantification of the approach localization resolution, localization accuracy, its dependence on the number of TOF measurements and exposure time, its sensitivity to ambient light, and measured the coherence time of the two light sources used.

In the response below, the referees comments are printed in *italics*, our answers are given in blue, and the changes to the manuscript in regular text.

In the revised manuscript, the changes and additions are highlighted in red.

We hope that you will find the revised manuscript appropriate for publication in *Nature Communications*.

Reviewers' comments:

Reviewer #1 (Remarks to the Author):

Recently, numerous studies pushed back the limits of non-line of sight(NLOS) imaging techniques. Usually, these techniques require pulsed sources and (very) highspeed photo-detectors in order to measure time of flight (ToF) of the order of picoseconds. This results in complex and expensive set up. In this article, the authors propose to adopt an approach already established in seismology, acoustics and more recently in optics to perform similar task, here source localization, but in a totally passive manner. Passive means no pulsed source or high speed-detectors are needed. This approach relies on the interferences of low coherence light emitted from the sources and measured with a conventional camera. Furthermore, the resolution is no longer dictated by the detectors but by the coherence time of the sources which can be in the femtosecond range for white light lamps.

I found this paper very interesting. It is well written and presents an elegant approach to simplify current techniques. The experimental setup is rather simple, cost-effective and the experimental demonstrations are convincing, especially as different configurations are explored. The supplementary information is also valuable. Maybe too much, as I found it really necessary to understand the approach (which is not that simple with the article only). In my opinion, this work is of interest for the community but the article in his current version lacks some clarifications on possible applications, more information on the experimental conditions and details on the limitations of this technique. Here is the list of my comments and concerns:

(1) First, the two advantages of the proposed technique are the passive nature of the measurement and its temporal resolution.

In NLOS imaging techniques, this temporal resolution is directly related to the spatial resolution by the celerity of light. A better temporal resolution, here the authors claim temporal resolution 3 order of magnitude better than conventional techniques using ultrafast detectors, should then provide higher spatial resolution. This is not the case in this article where the set up's geometry limits the spatial resolution as detailed in the supplementary information only.

It seems necessary to me to clearly describe what is the ideal spatial resolution (from the temporal resolution) and how the setup reduces it. Currently, the authors provide no number to characterize the spatial resolution of the system, they just mention "the object angular size", which is not practical at all.

For instance, I would like to know, how close can 2 sources be localized in the current configuration?

And more important, how can it be improved?

Right now, the results show sources separated by tens of centimeters, so way larger compared to conventional NLOS techniques where the spatial resolution seems to be in the millimeter-centimeter range. Can this method achieves similar performances? Or is it the best performances achievable with this technique?

We thank the referee for raising this point, which was not properly addressed in our original manuscript. In order to properly address the resolution limits of our approach, we have added both a theoretical estimate of the transverse (dx) and axial (depth, dz) resolution of our approach. In addition, we have performed a set of experiments with nearby sources in order to verify that the theoretical estimates are indeed achievable with the geometry of our setup. The results of these new analyses indeed prove that millimeter transverse resolution and centimeters-scale axial resolution are achievable with our setup. These new analyses are presented in the revised Supplementary Section 6, which we provide below.

To refer to these new data, as well as to the limitations and possible improvement of the localization resolution, we have added the following paragraph to our revised manuscript main text:

“Similar to other TOF-based NLOS imaging approaches, the localization resolution is determined by the geometry of the system and the TOF temporal measurement resolution. The transverse (dx) and axial (depth, dz) localization resolution from a single camera shot in our approach are analyzed theoretically in Supplementary Section 6.1. They are given by: $dx \approx l_c \cdot \frac{z}{D}$ and $dz \approx \frac{l_c \cdot z^2}{D(x+x_{slits})}$ (Supplementary Equations 8-9). Substituting our experimental parameters of $l_c = 6.3\mu m$, $x = 0$, $x_{slits} = 10mm$, and $z = 80cm$, yields $dz \approx 32cm$, and $dx \approx 4mm$, for the single-shot localization resolution. In order to verify our theoretical predictions for the localization resolution, we have performed a set of experiments with two sources separated by various transverse and axial distances. The results of these experiments, which validate our theoretical estimates, are presented in Supplementary Section 6.2.

...
Improved localization accuracy can be obtained by scanning the double-aperture mask over larger distances (x_{slits} in Supplementary Equation 8).“

The new and added sections to Supplementary Section 6 read:

“6.1 Theoretical estimate

Consider a distant object located at a distance z from the barrier, and at a transverse position x from the center of the measurement position. The light from this object arrives to the barrier at an angle of $\theta' = \text{atan}(x/z)$ measured in respect to the normal of the barrier (Supplementary Figure 6). The TOF difference in arrival time of the light from the object to the two measurement apertures having a separation D is:

$$\Delta t = \frac{\Delta L}{c} \approx \frac{D}{c} \sin(\theta') \quad \# (3)$$

This TOF difference is measured in our approach with a resolution that is given by the source coherence time: $\tau_c = l_c/c$, where l_c is the illumination coherence length. The angular localization resolution, $d\theta'$ can be estimated by

equating the differential of the measured time delay, Δt , to the measurement temporal resolution τ_c :

$$\tau_c = d(\Delta t) \approx \frac{D}{c} \cos(\theta') d\theta' \quad \#(4)$$

yielding:

$$d\theta' \approx \frac{l_c}{D \cos(\theta')} \quad \#(5)$$

Substituting our experimental parameters: $l_c = 6.3 \mu m$ (see Supplementary subsection 5.1) and $D = 1.25 mm$ at $\theta' \approx 0$, we obtain $d\theta' \approx 5 mrad$ for our experimental geometry.

The transverse (dx) and axial (dz) resolutions can be similarly derived by plugging into Supplementary Equation (3): $\sin(\theta') \approx (x + x_{slits})/z$, where x_{slits} is the center position of the moving slits, and taking the differential of Δt with respect to x and z separately, yielding:

$$d(\Delta t) \approx \frac{D}{c \cdot z} dx \quad \#(6)$$

and:

$$d(\Delta t) \approx \frac{D(x + x_{slits})}{c \cdot z^2} dz \quad \#(7)$$

Thus, the theoretically expected transverse and axial localization resolutions are given by:

$$dz \approx \frac{l_c}{D(x + x_{slits})} \cdot z^2 \quad \#(8)$$

$$dx \approx \frac{l_c}{D} \cdot z \quad \#(9)$$

Substituting our experimental parameters of $l_c = 6.3 \mu m$, $x = 0$, $x_{slits} = 10 mm$, and $z = 80 cm$ yields $dz \approx 32 cm$, and $dx \approx 4 mm$ for this distance.

6.2. Experimental localization resolution verification

To verify the theoretical predictions of the localization resolution we have performed a set of experiments with two white-light sources placed at various small axial and transverse separations from each other. The experimental results of the transverse localization resolution are presented in Supplementary Figure 7. Supplementary Figure 7a, displays the measured spatio-temporal (x - t) TOF trace for two sources positioned at $z = 60 cm$, separated by $\Delta x = 3 mm$, as measured with our optical setup and processed by spectrogram analysis (see Supplementary Section 3). At this distance, the theoretical transverse localization resolution (Supplementary Equation 9) is expected to be $dx \approx 3 mm$. Indeed, the two sources that are separated by this transverse spacing are resolved, but with a small contrast, as is also visible in Supplementary Figure 7b, where the cross-section of a single row of the trace of Supplementary Figure 7a is plotted.

We have repeated this experiment for different transverse separations $\Delta x = 3 - 12\text{mm}$, for two longer distances from the barrier: $z = 110\text{cm}$ and $z = 130\text{cm}$. The results of this study are presented in Supplementary Figure 7c,d. As expected from the theoretical estimation, sources separated by a transverse spacing of $\Delta x = 3\text{mm}$ are no longer resolved at the longer distances, and the individual sources are distinguished only for separations of $\Delta x = 6\text{mm}$ for $z = 110\text{cm}$ (Supplementary Figure 7c), and $\Delta x = 7\text{mm}$ for $z = 130\text{cm}$ (Supplementary Figure 7d), with good accordance with the theoretical estimations of Supplementary Equations 8.”

Supplementary Figure 7: **Transverse localization resolution, experimental results.** (a) Measured spatio-temporal traces for two light sources separated transversely by $\Delta x = 3\text{mm}$ at a distance of $z = 60\text{cm}$ behind the diffuser. (b) Envelope of the interference fringes for a single mask position taken from the cross-section marked by the dashed box in (a). (c) Same as (b) for two sources with various transverse separations, located at a depth of $z = 110\text{cm}$ from the barrier. (d) Same as (c) for two sources at $z = 130\text{cm}$. Scale bar: 92fs .

“The experimental results for axial localization resolution measured with our optical setup are summarized in Supplementary Figure 8. In each of these four experiments, two sources were present in the scene at the same transverse position, but at different depths. In all of these experiments one source is located at a fixed distance $z_1 = 110\text{cm}$ behind the diffuser, and the second source is located at a distance of $z_2 = 30\text{cm}, 45\text{cm}, 60\text{cm}, 80\text{cm}$ from the barrier respectively, with no transverse translation. The four traces show

the envelope of the interference fringes position on the camera as a function of the mask's position.

For the sources separated by $\Delta z = 50\text{cm}$ or larger (Supplementary Figure 8a-c) the two sources are clearly resolved at the top and bottom of the spatiotemporal traces, i.e. when the mask is shifted by the largest transverse distance, as expected from the theoretical prediction of Supplementary Equation 8.

However, the sources separated by $\Delta z = 30\text{cm}$ (Supplementary Figure 8d) are no longer resolved, in agreement with the theoretical prediction of Supplementary Equation 8. Verifying our theoretical analysis for the technique localization resolution.”

Supplementary Figure 8: **Axial localization resolution, experimental results.** (a-d) Envelope of the interference fringes position on the camera as a function of the position of the mask, for 2 light sources positioned at varying axial (depth) separations, $\Delta z = 30 - 80\text{cm}$ behind the diffuser, at the same transverse position. The depths of the two sources are marked by Z_1 and Z_2 above each trace. Scale bars: 92fs .

(2) My second question is also about the localization accuracy. Theoretically, 3 ToF measurements only are needed to estimate the position of a source. Here, more ToF are used which of course improve the accuracy (and improves the appearance of the experimental data). I think it could be interesting to display a plot of the accuracy as a function of the number of ToF.

We thank the referee for raising this important point, which we have not addressed in the original manuscript. Following the referee's suggestion, we have added a graph of the localization accuracy as a function of the number of ToF as a new Supplementary Section 7, which we provide below. We have

added the following statement to the revised manuscript main text in order to refer to the localization accuracy:

“The final localization accuracy is considerably better than the single-shot localization resolution, since it is obtained from multiple TOF measurements. An experimental study of the localization accuracy as a function of the number of TOF measurements is presented in Supplementary Section 7.”

The new Supplementary Section 7 reads:

7. Localization accuracy

As mentioned in the main text, the localization accuracy is significantly better than the localization resolution, since it is obtained from multiple, high-SNR measurements. In order to study the transverse and axial localization accuracy in our experiments as a function of the number of TOF measurements taken (number of different mask's positions), we have acquired TOF data for a single source located at various transverse positions at a distance of $z = 57\text{cm}$ behind the diffusive barrier. For each source position, we have localized the source from a different number of TOF measurements ranging from 2 to 42. From each TOF measurement a plot of the corresponding hyperbola was back-projected (Supplementary Figure 9a). The point of intersection of the back-projection hyperbolas was found by convoluting the hyperbolas with a spatial blurring kernel of size 40 by 40 pixels in the reconstruction grid (0.4mm along the vertical axis (x) and 4.8cm along the horizontal axis (z) of Supplementary Figure 9a) and marking the peak of the resulting trace as the estimated position of the source. These positions together with the back-projected hyperbolas are presented in Supplementary Figure 9a: The reconstructed positions are marked by blue 'x', and the real positions of the source are marked by white 'x'.

The localization accuracy was estimated by calculating the standard deviation of the estimated depths (Δz) and the transverse (Δx) positions, from the real positions of the sources. The results for the localization accuracy in depth and transverse position as a function of the number of TOF measurements (number of back-projected hyperbolas) are presented in Supplementary Figure 9b,c, respectively. As expected, the localization accuracy is improved when more TOF measurements are used. At this depth ($z = 57\text{cm}$) the transverse localization accuracy with our experimental parameters reaches $\Delta x = 0.2\text{mm}$, and the depth resolution reaches $\Delta z = 1\text{cm}$. The transverse resolution is two orders of magnitude better than the axial resolution in our experiments, as expected from the localization resolution (see Supplementary Section 6).

Supplementary Figure 9: **Localization accuracy of a single hidden source as a function of the number of TOF measurements (mask positions).** (a) Plot of back-projected hyperbolas from all TOF measurements for a single source placed at several transverse positions at a distance $z = 57\text{cm}$ behind the diffuser. The intersections of the hyperbolas reveal the different reconstructed positions (marked by blue 'x'), using a different number of TOF measurements. The real positions are marked by white 'x'. (b) Axial localization accuracy as a function of the number of TOF measurements, as calculated from the standard-deviation of the reconstructed positions in (a). (c) Same as (b) for the transverse localization accuracy.

Referring the referee statement on the minimal number of measurements required, we wish to note that while the referee is correct that only three ToF measurements are required for 3D localization of a single source. For a scene containing multiple nearby sources more measurements are needed in order to unambiguously localize the sources. To make this point clear we have added the following clarifying statement to the revised manuscript:

“As in active TOF based NLOS imaging approaches, localizing multiple hidden sources without ambiguity requires a larger number of TOF measurements.”

(3) The authors mentioned only once the exposure time. Are the values (few seconds for the diffusive barrier and 15 minutes for the around the corner experiment) for one ToF measurement only or for the total acquisition? I think it should be mentioned for each experiment.

Following the referee's suggestion, we mention the single-shot exposure times and total acquisition time in the revised manuscript for each experiment. We also clearly note that the mentioned exposure time is for a single ToF

measurement, i.e. one row in the spatio-temporal traces. The following paragraph was added to the Discussion section:

“In our implementation, integration times of a few seconds for a single camera shot were required to provide sufficiently high contrast fringes from the white light LED sources in the diffusive barrier case (Fig.3), under dark conditions. In the experiments with reflective objects (Fig.5), exposure times of 30 seconds were used for a single exposure. In the reflective white-painted wall experiments, exposure times of ~15 minutes per camera shot were required (Fig. 4). These single-shot exposure times yielded acquisition times of minutes to hours for the total number of measurements presented. Lower exposure times lead to lower fringe visibility and signal to noise ratio (SNR) but can still be useful for TOF measurements, as we study in Supplementary Section 8. A lower number of measurements can be used for localization at a lower accuracy (see Supplementary Figure 9).”

(4) Also, what is the acquisition scheme? Is it a single frame with long exposure or multiple images with low exposure averaged in post processing?

We thank the referee for pointing this important missing information. The acquisition scheme is a single frame with long exposure. This is now explained in the revised Materials and Methods section:

“Each TOF measurement was acquired by a single long-exposure (15 seconds to 15 minutes) image”

I think adding a convergence curve (visibility of fringes as a function of exposure time) and examples of images with different exposure time would be really useful.

Following the referee’s suggestion, we have analyzed the visibility of the fringes amplitude as a function of the exposure time and have added the results of this experimental study to a new Supplementary Section 8, which we append below. This section is referred to in the revised manuscript main text by the following statement:

“Lower exposure times lead to lower fringe visibility and signal to noise ratio (SNR) but can still be useful for TOF measurements, as we study in Supplementary Section 8.”

The new Supplementary Section 8 reads:

"8. Effects of ambient light and exposure time

In this section we experimentally study the effects of ambient lighting, background reflections, exposure time, and the measured fringe contrast.

Similar to OCT, beyond the absolute signal level, one of the main challenges of our passive TOF technique is the signal to background ratio, or the fringe contrast. The experimental results presented in Figures 2-5 were performed under dark conditions. In order to study the practical effects of ambient light illumination and white reflective background, we have performed a set of measurements whose results are summarized in Supplementary Figure 10. Supplementary Figure 10 shows an analysis of the fringes envelope SNR, under different illumination conditions and exposure times. The SNR was quantified by taking the amplitude ratio between the peak of the fringes envelope and the standard deviation of the background signal in a single camera shot (a single row in Supplementary Figure 10a-c). Three scenarios were considered: the first is a dark conditions scenario (Supplementary Figure 10a,d), the second is a scene with ambient room light (Supplementary Figure 10b,e), and the third is under ambient room light in addition to a white painted cardboard placed 20 cm behind the hidden target (Supplementary Figure 10c,f). The results of these experiments are summarized in Supplementary Figure 10g. As expected, the presence of ambient light, a reflective background, and shorter exposure times, all lower the fringe contrast as quantified by the SNR. However, localization of small white-light LED sources is possible even under ambient light conditions."

Supplementary Figure 10: **Effect of ambient illumination, background reflection, and exposure time on experimentally-measured fringe contrast.** (a-c) Examples spatio-temporal traces for a single source in three different scenarios: (a) a dark room, (b) with ambient room light, and (c) with ambient room-light and a white-painted cardboard located 20cm behind the LED light source. Exposure time per row is 30sec. (d-f) cross sections of a single row in (a-c), for different exposure times. (g) Calculated signal to noise ratio (SNR) of the fringes as a function of the exposure time for the measurements shown in (d-f). Scale bars: 92fs.

(5) The authors state the cross correlation is sampled at thousands of different time delays (line 111). Can they be more specific? The camera has 5.5 Mpixels, so do they use the full resolution? Maybe the authors can state the number of sampling points is given by the number of pixels (if I understood well), and then give the number of pixels used in their experimental conditions.

This is correct. The number of time delays sampled in a single measurement is limited by the number of camera pixels in one camera row. To clarify this point we have added the following clarification to the manuscript main text:

“The number of time delays that can be sampled in a single camera image is limited by the number of pixels in a single camera row, which was 2160 in our experiments. “

In addition, we have added the following paragraph to supplementary subsection 5.1:

“The number of time delays that can be sampled in a single camera image is limited by the number of pixels in a single camera row. In our experiments this number was limited to 2,160 pixels by the camera pixel count, and to 1400 pixels by the specific optics used, which limited the angular field-of-view of the speckles collected on the camera (the diameter of the speckled circle in Fig. 2b).”

(6) Are the experiments performed with a background light? Is this approach doable outside for instance?

The experiments presented in our original manuscript were performed under dark conditions. To address the referee’s important question regarding the feasibility of the approach under ambient light conditions, we have performed an experimental study of the fringe visibility as a function of exposure time, with and without ambient light illumination, including the presence of a white-painted background reflecting wall placed behind the hidden LED source. These new experimental results are presented in the completely new Supplementary Section 8. In addition, in order to address the background lighting conditions in the main text, we have added the following paragraph to the revised manuscript’s main text:

“To achieve a sufficiently high fringe contrast, the experiments presented in Figures 2-5 were performed under dark conditions. The effects of ambient light and bright reflecting background on the fringe contrast are studied in Supplementary Section 8. For the hidden white-light LED sources similar localization results can be obtained under both dark and ambient-light conditions. For the reflective objects case, the quality of the results depends on the relative illumination intensity of the objects compared to the background. The requirement for bright sources or reflectors over a relatively

dark background is a fundamental requirement for obtaining high contrast fringes. However, as in OCT, low contrast fringes may be resolved by longer exposure times, achieved by averaging multiple exposures or by large well-depth cameras.”

The new Supplementary Section 8 appears in our reply to point (5) above

(7) What is the transmission efficiency of the diffuser and of the mask?

The transmission of the diffuser is larger than 90% at visible wavelengths. The holes in the mask provide 100% transmission. We have revised the following sentence in the Materials and Methods section to include these details:

“The diffusive barrier was a Newport light shaping diffuser with a scattering angle of 80 degrees and transmission of > 90% in the visible range.”

(8) Why the authors choose this particular mask? With only 2 groups of 2 square elements? Why the authors did not choose rounded apertures for instance? And why not groups of 3 apertures (in a triangle form) to provide both x and z at the same time?

In our proof of principle experiments, we wished to use the simplest masks that would be the most straightforward to understand and analyze. These masks demonstrated multiplexing of two TOF measurements in a single camera shot. The referee is perfectly correct that more complex mask designs can be used for improved multiplexing, as we have mentioned in our original manuscript:

“Reduced acquisition times and number of acquisition may be possible by using multiplexed detection using masks with a large number of apertures [P.G. Tuthill (2000)], and by using more advanced reconstruction algorithms.”

To further clarify this point, we have added the following statement to the revised manuscript:

“In our proof-of-principle experiments we have chosen to use simple masks comprised of two groups of circular-apertures pairs, to multiplex two TOF measurements in a single camera exposure. More complex masks can be used to allow scene reconstruction from a single camera exposure, in a similar manner as is done in aperture masking interferometry in astronomy apertures [P.G. Tuthill (2000)].”

Regarding the question on the shape of the apertures: we have used round apertures in all of our mask-based experiments and have used rectangular apertures only in the SLM-based experiments, due to the square dimensions

of the SLM pixels. We apologize for the misleading mask sketches depicting rectangular apertures that were presented in the original manuscript. In the revised manuscript we have corrected all of the relevant figures such that they display circular apertures in all of the mask-based experiments, e.g.:

To clarify this point, we have mentioned the circular apertures shape in the statement presented above, and have also added the following statement to the revised manuscript Supplementary Section 1:

“Circular apertures were used in our mask-based experiments, and rectangular apertures were used only in the SLM-based experiments, due to the square dimensions of the SLM pixels.”

(9) In the experiment around the corner, the ToF results display both a broader and a rougher response. From the broadening, is it possible to estimate the scattering properties of the wall? Does the rough aspect comes from the rough surface of the wall or just from lower signal to noise ratio? would it be possible to estimate other parameters on the wall from the roughness of the ToF?

We thank the referee for raising this point, which was only mentioned in passing in the original manuscript. Indeed, the rougher response of the ToF measurements from the white-painted wall originates from the wall roughness, and the broader response is a combined result of the surface roughness and the multiple-scattering nature of the white-painted wall. In order to properly address this point, we have added a quantification of the broadening and roughness of the TOF curves for the two barriers used in our work, and have added the following explanation to the revised manuscript:

“As can be observed in Figure 4c, the TOF traces from the reflective wall display TOF fluctuations that are larger than those observed through the optical diffuser (Figure 3b). This effect is a result of the rough nature of the white-painted wall. It can be observed and quantified thanks to the unique femtosecond-scale temporal resolution of our approach, whereas picosecond pulses used in previous works would mask these effects. To show that the barrier roughness indeed induces measurable variations in the TOF with our approach, we calculated the standard-deviations of the measured TOF for both the diffusive barrier in transmission and the white-painted wall in reflection to be ~ 2.3 fs, and ~ 10 fs, respectively (see Supplementary Figure 12).

The surface roughness and the multiple-scattering nature of the reflections from the white-painted barrier, induce not only fluctuations in the TOF differences, but also a temporally broader impulse response. This temporal broadening is also measurable in our approach, as is shown in Figure 4e. This figure presents a comparison of the fringes envelope as a function of the time delay for the case of a thin diffuser vs. the white painted barrier. Compared to the diffuser case, for the white painted wall we observe a main peak that is broader by a factor of ~ 2 , accompanied by a broader pedestal. This broadening, in addition to the TOF fluctuations due to the surface roughness, effectively lower the TOF resolution and reduces the localization accuracy. Nonetheless, the TOF resolution is still better than using picosecond pulses in our proof-of-principle experiments. Interestingly, the changes in the TOF curves that are induced due the nature of the barrier, and are resolvable with our approach, carry information on the barrier. For sufficiently small point-like hidden sources, surface and scattering properties of the barrier, such as its transport mean free path, can in principle be estimated from the temporal broadening of the low-coherence fringes envelope [Badon et al. (2015)].”

The quantitative comparison between the TOF traces through the diffuser and in reflection from the white-painted wall is presented in a new Supplementary Section (10) in the revised manuscript. The main figure in this section reads:

Supplementary Figure 12: **Comparison of TOF traces obtained through a diffuser and from white-painted surface scattering.** (a-b) Traces of the interference fringes envelope as a function of the mask position for localization through a diffuser (a) and for light reflected from white-painted surface (b). (c) Cross sections of a single row in (a) and (b). Scale bars: $92fs$.

(10) The two lamps used in the experiments have different spectral bandwidths. Do the authors noticed different temporal resolutions in the experiments conducted with these two light sources?

Indeed, the two sources have different spectra, and thus their spectral bandwidths yield autocorrelation traces with slightly different temporal resolutions. To properly address this point, we have added a measurement of the spectra of the two light sources and calculated their respective field autocorrelation by Fourier transforming the measured spectra. We have compared these calculated values to the values of the TOF resolution of our experiments with a diffusive barrier. The results of these new measurements are presented in the new Supplementary Section 5.1, which reads:

5.1. TOF resolution of the different sources used

The temporal TOF resolution of our approach is dictated by the source coherence time, i.e. the envelope of the source field autocorrelation. Which, according to the Wiener-Khinchin theorem, is given by the Fourier transform of the source power spectrum. In our experiments we have used two light sources: a white light LED and a Tungsten-halogen lamp (Thorlabs OSL2). These two sources have different spectra (Supplementary Figure 5a,d), and thus the autocorrelation envelope of these two sources, as calculated by Fourier-transforming their spectra, have slightly different temporal widths (Supplementary Figure 5b,e).

Specifically, the full width half maximum (FWHM) of the of the field autocorrelation envelope is $\tau_{FWHM} \approx 6.6fs$ for the LED source, and $\tau_{FWHM} \approx 6.1fs$ for the Halogen lamp (Supplementary Figure 5c,f).

These resolutions are three orders of magnitude better than ultrafast detectors used in conventional TOF techniques, which are of the order of 15ps using a streak camera [Velten et al. (2012)], and 8ps using SPAD detectors [O'Toole et al. (2018)].

Supplementary Figure 5: **Spectra and resulting auto-correlation of the white light sources used in our experiments.** (a) White LED source measured spectrum. (b) Absolute value of the field autocorrelation calculated by a Fourier transform of the spectrum in (a). (c) Measured interference fringes envelope as detected on the camera using the LED source in our experiments through a diffusive barrier. (d-f) Same as (a-c) for the Halogen lamp used in the experiments of Fig. 5.

In addition to calculating the coherence time of each of the sources, we have compared the TOF resolution obtained with the two sources in our experiments. This was done by comparing the envelope of the interference fringes (as detected with a spectrogram analysis) obtained in the localization

experiments with the two sources (Fig.3 and Fig.5) for a single source/object. Taking the FWHM of the envelope gives an estimate of the experimental TOF resolution of $\tau_{FWHM} \approx 21fs$ for the LED source (Supplementary Figure 5d) and $\tau_{FWHM} \approx 17fs$ for the Tungsten-Halogen lamp. The TOF resolution in experiments is lower than the coherence-time of the sources due to the presence of the diffusive barrier, optical aberrations of the setup, and the effects of windowing of the spectrogram analysis."

Minor comments

a) *The references line 22 should be splited in 2 and ordered by date.*

We have divided the references to two and ordered by date each group of references related to a single subject.

b) *Line 235 : there is a comma after star*

We corrected the type in line 235.

Reviewer #2 (Remarks to the Author):

This paper describes a technique for localizing a single or a few points through a thin diffuser or "around a corner". What's unique about this approach is that, unlike recent work in this important and timely research area, it does not rely on pulsed illumination and ultra-fast detectors nor does it require coherent illumination; it is therefore a passive imaging approach to non-line-of-sight imaging. Several other passive techniques to this problem have been proposed over the last few years, as adequately discussed in the manuscript, but the proposed method uses interference created by a "coded aperture" (pairs of horizontal or vertical points that are mechanically scanned on the diffuser plane) to achieve substantially higher resolution for the localization problem.

The idea behind the proposed technique is intuitive and closely related to Young's double slit experiment: the light emitted or reflected by a point source is constrained by only allowing it to pass through two small points or slits. After propagating by some distance, an interference pattern is created on the sensor. Point sources emitting light from different directions on the points/slits create a phase shift on the slit plane that is proportional to the incident direction, which in turn shifts the interference pattern. While a single image of the interference pattern is insufficient to localize the direction of the source, two approaches to using this effect for non-line-of-sight imaging seem possible. First, one could track a moving point by observing the shift of the interference pattern (not discussed in this paper) or (2) one could vary the distance between the slits for a stationary point source (proposed in this manuscript). This authors introduce the latter concept for the application of non-line-of-sight imaging and experimentally validate it for several conditions, including a single and multiple point sources, points behind a diffuser, and points scattered off of a reflective wall.

Overall, this is a very clever and intuitive technique, the manuscript is well written, and the experimental results are convincing in that they demonstrate that this intuitive concept is indeed applicable to source localization in non-line-of-sight imaging scenarios.

We sincerely thank the referee for his positive view of our work.

Reviewer #3 (Remarks to the Author):

The paper describes a method for localizing light sources obscured by diffusers using an interferometric technique. The paper is well written and does a good job exploring different applications of this method. I have only a few comments:

1. Reference [30] seems to describe a related method, but is only ever cited in passing. A more detailed comparison to this work in the introduction is warranted here as it seems on first sight to be the work that is closest related to this manuscript.

To address this point, we have added the citation of reference [30] ([Liu et al. 2015]), and the following short discussion, to the introduction in the revised manuscript. As a result, this reference is now numbered [17]:

“Phase-space based approaches [Liu et al. 2015], which localize hidden sources by measuring the positions and angles of the scattered light, suffer as well from a localization resolution that is dictated by the scattering angle of the barrier. Large scattering angles blur the k-space traces for objects that are not adjacent to the barrier.

A considerably higher resolution that is insensitive to the scattering angle of the barrier was recently obtained using speckle correlations [...] or by time-of-flight (TOF) based approaches...”

...

“Unlike intensity-only [Klein et al (2016), Bouman et al. (2017)] and phase-space measurements [Liu et al. (2015)] approaches, the localization resolution in our approach is dictated by the TOF temporal resolution and not by the scattering angle of the barrier. We thus present localization results obtained through highly scattering diffusers, having a scattering angle of 80° , and using light scattered off white-painted barriers. “

Indeed, while the optical setup of Liu et al. contains similar elements to those used in our work, the exact configuration and working principle are completely different: In the work of Liu et al. the camera is placed in the imaging plane of one target plane, and an SLM is placed at the Fourier plane to allow “phase-space” measurements. These measurements are simply multiple images of the target plane taken using angular filtering at different angles. The results are blurred images of the targets that are ideally shifted in position as the k-space angle is changed, which is essentially the same effect used in parallax-based localization. The technique assumes the position shift is measurable even through a scattering barrier. However, since the phase-space curves are blurred in k-space (angle) and in real-space by the scattering angle of the diffuser (see example below), and the angular spread of the source illumination angle, the localization resolution of phase-space approach is inherently limited by the scattering angle of the diffuser, and is expected to be

poor through the highly scattering (80 degrees) diffusers and diffusive walls considered in our approach. An example for such a spread, taken from reference [30] is given below:

Since our approach is based on accurate ToF measurement it is insensitive to the scattering angle of the diffusive barrier (assuming all the other parameters remain the same) – i.e. the spatio-temporal curves measured in our approach (and other TOF techniques) would be the same with and without the scatterer, except for the small blurring in TOF due to the roughness of the barrier (see our reply to question 9 of Referee 1 above). This is very different from the results of reference [30] shown above.

2. In computational imaging "passive" techniques are usually those that work with ambient light, while "active" techniques require control of the illumination, for example from a projector. Using this definition, the presented method is active since it requires the targets to light up and presumably does not tolerate ambient light. Even the setup described in Figure 5(a) apparently uses illumination that is aimed only at the targets. I would recommend replacing the term Passive by something less confusing. Maybe 'coherence gated', 'interferometric', or 'time-of-flight with conventional cameras'

We thank the referee for raising this point. As noted by referee 1 and 2, we referred to the technique as 'passive' in the sense that it requires no pulsed source or high-speed detector, and that it can work with natural incoherent

light. We agree that, unlike active TOF techniques, for our technique to work, i.e. for being able to resolve the low-coherence fringes, the targets have to be bright enough compared to the background. This is a limitation shared with other low-coherence based approaches, such as OCT, where low-contrast fringes need to be detected over a strong background.

In order to address this important point and to study the effects of ambient light conditions on our measurements we have performed a set of experiments whose results are summarized in the new Supplementary Section 8. This new Supplementary Section appears in our reply to question 4 of referee 1 above. These results suggest that for sufficiently bright targets ambient light conditions can be tolerated by our approach.

We agree that further study beyond our proof-of-principle experiments is required to assess whether natural reflective targets can be localized under daylight outdoors environments with our approach. However, it is important to note that, as in OCT, using longer exposure times and higher well-depth cameras, as well as other more advanced approaches, can be used for resolving extremely low contrast interference fringes under more difficult illumination conditions.

To address this important point raised by the referee, we have added the following statement to the revised manuscript:

“The requirement for bright sources or reflectors over a relatively dark background is a fundamental requirement for obtaining high contrast fringes. However, as in OCT, low contrast fringes may be resolved by longer exposure times, achieved by averaging multiple exposures or by large well-depth cameras.”

Since our approach can in principle allow passive TOF based localization, we wish to keep the current title. We are open to changes in the title, as long as they do not present an additional confusion: We believe that adding ‘interferometric’, ‘coherence gated’, or ‘conventional cameras’ to the title may confuse the reader to think that we present an OCT based approach, where the source can be a broadband laser that is completely controlled and operated at the measurement system side. We believe that the main uniqueness of our approach is that the sources are natural light sources that are located at the *targets side* behind the barrier. Thus, we used the term ‘passive’ in the title, but we would be very happy to consider a different change in the manuscript title.

Overall I recommend publication of the manuscript.

We sincerely thank the referee for his positive view of our work.

Reviewer #2 (Remarks to the Author):

After reading the other reviews, the point by point response of the authors, and the revised manuscript and supplement, I still think that the work is generally clever and also unique in the area of non-line-of-sight localization. Reviewer 1 brought up a lot of important details, which I believe were adequately addressed by the authors in this revision. The detailed discussion and experimental validation of localization resolution and precision along with many other additional details and clarifications certainly improved the submission.

However, I am somewhat concerned about the general positioning of the capabilities of this work. In the abstract and throughout the manuscript, it is argued that the proposed method has fs resolution and thereby surpasses state-of-the-art approaches that use ps-accurate detectors by orders of magnitude. In principle, all of this is certainly true, yet the experimental results are very limited and only show one or a few emitters or reflectors whereas previous work has demonstrated NLOS imaging of significantly more complex 3D scenes. I'm not too concerned about the quality of the presented results - the authors probably report results that this method is capable of achieving in general, so I'm not asking for new results or experiments. All of this is of course captured in the subtle distinction between of phrasing it as a "localization" rather than a "3D reconstruction / imaging" approach, so the work is not misrepresented. But the discussion in the paper is misleading in creating the impression that the method generally greatly outperforms existing TOF-based NLOS approaches, which I certainly do not believe to be the case. A more balanced discussion of the benefits and disadvantages of each of these methods is imperative and I would feel uncomfortable to accept the manuscript without a more balanced discussion. I don't think it would be sufficient to add a sentence or two at the end of the paper, but believe it is required to adequately position this work w.r.t. prior work throughout the manuscript. Yes, the proposed method is able to localize a single or very few emitters/reflectors much more accurately than other methods and, yes, a passive setup is less expensive than one that requires ps pulsed lasers. But does that automatically lead to a more practical approach to NLOS imaging or better results for complex scenes? I don't think it does, because the required exposure times are long and the method is fundamentally limited in the complexity of NLOS scenes it can handle. Similar to Young's well-known double slit experiment, which again is very similar to this work, I see this work more as a thought experiment with proof-of-principle demonstration than a practical NLOS imaging approach. Does that mean the paper should not get published? I don't think so, as there is high value in thought experiments as they may help readers intuitively understand this problem along with fundamental limits of resolution and precision of NLOS imaging methods, and stimulate new ideas to push these limits. But an adequate positioning of the work seems necessary to avoid misinterpreting this work by a non-expert reader.

Reviewer #3 (Remarks to the Author):

My concerns have been addressed. Given the feedback of the authors and the other reviewers, I am okay with keeping the word passive in the title.

I think this is an important advance to the field and I recommend publication.

Please find attached below our point-by-point response to the each of the referees comments. We sincerely thank the referees for their insightful comments and suggestions that allowed us to significantly improve our manuscript, and to adequately position our work with respect to prior works.

In the response below, the referees comments are printed in *italics*, our answers are given in blue, and the changes to the manuscript in regular text.

In the revised manuscript, we have implemented all of the changes requested by the referees, have corrected a few typos, and have clarified a few phrasing. The changes to the revised manuscript are highlighted in red in the revised text.

We hope that you will find the revised manuscript appropriate for publication in *Nature Communications*.

Reviewers' comments:

Reviewer #2 (Remarks to the Author):

After reading the other reviews, the point by point response of the authors, and the revised manuscript and supplement, I still think that the work is generally clever and also unique in the area of non-line-of-sight localization. Reviewer 1 brought up a lot of important details, which I believe were adequately addressed by the authors in this revision. The detailed discussion and experimental validation of localization resolution and precision along with many other additional details and clarifications certainly improved the submission.

However, I am somewhat concerned about the general positioning of the capabilities of this work. In the abstract and throughout the manuscript, it is argued that the proposed method has fs resolution and thereby surpasses state-of-the-art approaches that use ps-accurate detectors by orders of magnitude. In principle, all of this is certainly true, yet the experimental results are very limited and only show one or a few emitters or reflectors whereas previous work has demonstrated NLOS imaging of significantly more complex 3D scenes. I'm not too concerned about the quality of the presented results - the authors probably report results that this method is capable of achieving in general, so I'm not asking for new results or experiments. All of this is of course captured in the subtle distinction between of phrasing it as a "localization" rather than a "3D reconstruction / imaging" approach, so the work is not misrepresented. But the discussion in the paper is misleading in creating the impression that the method generally greatly outperforms existing TOF-based NLOS approaches, which I certainly do not believe to be the case. A more balanced discussion of the benefits and disadvantages of each of these methods is imperative and I would feel uncomfortable to accept the manuscript without a more balanced discussion. I don't think it would be sufficient to add a sentence or two at the end of the paper, but believe it is required to adequately position this work w.r.t. prior work throughout the manuscript. Yes, the proposed method is able to localize a single or very few emitters/reflectors much more accurately than other methods and, yes, a passive setup is less expensive than one that requires ps pulsed lasers. But does that automatically lead to a more practical approach to NLOS imaging or better results for complex scenes? I don't think it does, because the required exposure times are long and the method is fundamentally limited in the complexity of NLOS scenes it can handle. Similar to Young's well-known double slit experiment, which again is very similar to this work, I see this work more as a thought experiment with proof-of-principle demonstration than a practical NLOS imaging approach. Does that mean the paper should not get published? I don't think so, as there is high value in thought experiments as they may help readers intuitively understand this problem along with fundamental limits of resolution and precision of NLOS imaging methods, and stimulate new ideas to push these limits. But an adequate positioning of the work seems necessary to avoid misinterpreting this work by a non-expert reader.

We sincerely thank the referee for raising this point. Indeed, after carefully reading again the previous version of our manuscript, we agree that a non-expert reader might have gotten the impression that the proposed approach is superior to conventional active TOF approaches, while this is indeed true only from the point of view of localization resolution, accuracy, and passive-nature, and only when sparse scenes are considered.

In order to properly correct this impression in the revised manuscript, we have followed the referee request and have mentioned the disadvantages and limitations

of our technique at every instance throughout the manuscript where it is compared to conventional TOF technique. To make this point perfectly clear, we have also added statements regarding the limitation of the approach **already in the abstract** and introduction section, in addition to the requested revisions in the discussion section.

The revised concluding lines of the abstract have been changed, such that the word 'imaging' does not appear in the abstract, and the fundamental limitations of our approach are clearly stated. These lines read:

“

Our approach retrieves TOF information from temporal cross-correlations of scattered light, **via interferometry**, providing temporal resolution that surpasses state-of-the-art ultrafast detectors by three orders of magnitude.

While our passive approach is limited by signal-to-noise to relatively sparse scenes, we demonstrate passive localization of multiple white-light sources and reflective objects hidden from view using a simple setup.

“

The following changes have been made to the manuscript text:

Introduction: The concluding paragraph in the introduction have been changed to read:

“

However, while the localization resolution of our approach is superior to what is possible with ultrafast detectors, due to the passive nature of our approach and its reliance on low-coherence interferometry, it is fundamentally limited by signal-to-noise (SNR) considerations to the localization of a relatively small number of sources or reflectors, and it requires relatively long exposure times.

“

Discussion:

The following sentences were added to the first paragraph of the discussion section:

“

However, as we discuss in detail below, while the localization resolution is better than active techniques, as we explain below, the passive nature of our approach and its low-coherence interferometry based measurements fundamentally limit its use to localizing a relatively small number of small sources or reflectors, a limitation that does not occur in active TOF based approaches, where the measurement background is low compared to the active pulsed illumination intensity

“

The limitations of our technique are already analyzed in detail in the later parts of the previously revised Discussion section. These parts read:

“The passive nature of our TOF approach is also its main disadvantage: since light from natural light sources is considered, the detected intensity levels are inherently low...

These single-shot exposure times yielded total acquisition times of minutes to hours for the total number of measurements presented.

“

“

To achieve a sufficiently high fringe-contrast, the experiments presented in Figs. 2-5 were performed under dark conditions. The effects of ambient light and bright reflecting background on the fringe contrast are studied in Supplementary Note 8.

“

“

The requirement for bright sources or reflectors over a relatively dark background is a fundamental requirement for obtaining high contrast fringes.

“

“

However when extended objects are concerned, the aperture separation must be smaller than the source coherence size on the barrier, r_{coh} , to obtain high contrast fringes. This limits the apertures separation when large extended sources or objects are considered, and limits the single-shot localization resolution to the source dimensions (see Supplementary Note 6).

“

We believe that with the above revisions the revised manuscript now adequately positions our approach with respect to prior works, such that the work would not be misinterpreted by a non-expert reader. We wish to thank again the referee for highlighting this important point.

Reviewer #3 (Remarks to the Author):

My concerns have been addressed. Given the feedback of the authors and the other reviewers, I am okay with keeping the word passive in the title.

I think this is an important advance to the field and I recommend publication.

We thank the referee for his positive view of our work.

REVIEWERS' COMMENTS:

Reviewer #2 (Remarks to the Author):

All my remaining concerns have been adequately addressed in this revision.

Please find attached below our response to the referees. We again wish to express our gratitude to the referees for their insightful comments and suggestions that allowed us to significantly improve our manuscript.

In the response below, the referees comments are printed in *italics*, our answers are given in blue.

Reviewers' comments:

Reviewer #2 (Remarks to the Author):

All my remaining concerns have been adequately addressed in this revision.

We sincerely thank the referees for the thorough and important discussion.